# Outcomes of Acute Kidney Injury Among Hospitalized Patients with Sepsis and Acute Myeloid Leukemia: A National Inpatient Sample Analysis

**DOI:** 10.3390/jcm14072243

**Published:** 2025-03-25

**Authors:** Hari Naga Garapati, Deepak Chandramohan, Boney Lapsiwala, Udit Nangia, Devansh Patel, Prabhat Singh, Sreekant Avula, Aditya Chauhan, Nihar Jena, Prathap Kumar Simhadri

**Affiliations:** 1Department of Internal Medicine/Nephrology, Baptist Medical Center South, Montgomery, AL 36116, USA; drvenug@gmail.com; 2Department of Internal Medicine/Nephrology, University of Alabama at Birmingham, Birmingham, AL 35233, USA; deepakchandramohan@uabmc.edu (D.C.); devanshhimanshupatel@uabmc.edu (D.P.); 3Department of Internal Medicine, HCA Medical City Healthcare, Arlington, TX 76015, USA; boney0104@gmail.com; 4Department of Internal Medicine, University Hospitals-Parma Medical Center, Parma, OH 44129, USA; udit.nangia2@gmail.com; 5Department of Internal Medicine/Nephrology, Kidney Specialists of South Texas, Corpus Christi, TX 78404, USA; drprabhatsingh@hotmail.com; 6Department of Internal Medicine/Endocrinology, University of Minnesota Twin Cities, Minneapolis, MN 55455, USA; chauh111@umn.edu; 7Department of Internal Medicine/Cardiovascular Medicine, Wayne State University, Detroit, MI 48201, USA; niharjmd@gmail.com; 8Department of Nephrology, Advent Health/FSU College of Medicine, Daytona Beach, FL 32117, USA; prathap.simhadri@gmail.com

**Keywords:** acute kidney injury (AKI), acute myeloid leukemia (AML), sepsis, mortality

## Abstract

**Background:** Sepsis in patients with acute myeloid leukemia (AML) is one of the causes of acute kidney injury (AKI). There are no available data on the outcomes of AML-related AKI patients. **Methods:** We researched the 2016–2020 National Inpatient Sample (NIS) database to collect data on hospitalizations of patients ≥18 years old with sepsis and AML. These admissions were divided into two weighted groups, with and without AKI. A multivariable logistic regression was used with adjustment for possible confounders to generate the adjusted odds ratios for the outcomes of the study. A *p*-value of <0.05 was considered significant. The primary outcome was all-cause inpatient mortality. Secondary outcomes were septic shock, fluid and electrolyte disorders, length of stay (LOS), vasopressor support, and the requirement for mechanical ventilation. **Results:** Out of 288,435 hospital admissions of patients with sepsis and AML, 61,955 (21.4%) had AKI. Patients with AKI were older (mean age 66.1 vs. 60.4 years), males (63.1% vs. 52.8%), and more Black individuals were affected (12% vs. 9.2). They also had more comorbidities but had a significantly higher percentage of diabetes mellitus, congestive heart failure, cardiac arrhythmias, cerebrovascular disease, and chronic kidney disease. Tumor lysis syndrome was present in 11.1%. Compared to patients without AKI, patients with AKI had longer LOS days (15.4 ± 18 vs. 10.8 ± 13.1, *p* < 0.001. Multivariable analysis showed that the patients with AKI had higher odds of mortality (OR: 3.8, 95% CI: 3.6–4.1, *p* < 0.001). They also had a higher risk for fluid and electrolyte disorders (OR: 2.2, 95% CI: 2.1–2.4, *p* < 0.001), septic shock (OR: 6.3, 95% CI: 5.7–6.9, *p* < 0.001), vasopressor requirement (OR: 5.0, 95% CI: 4.3–5.8, *p* < 0.001), and mechanical ventilation (OR: 5.2, 95% CI: 4.7–5.7, *p* < 0.001). **Conclusions:** AKI in patients with sepsis and AML was associated with higher mortality compared to sepsis alone, as well as other complications. Further large studies are required to identify factors that could improve outcomes.

## 1. Introduction

Acute myeloid leukemia (AML) is the most common acute leukemia in adults, accounting for approximately 25% of all leukemias among adults [1]. The incidence of AML increases with age, from approximately 1.3 cases per 100,000 in patients under 65 years old to 12.2 cases per 100,000 population in those over 65 years [2]. Recent advances in the treatment of AML have led to significant improvements in younger patients, but mortality remains high in older patients, who account for most new cases [2,3].

Patients with AML have a high rate of hospitalizations compared to those with other cancers (16% vs. 4%), and a considerable number of these hospitalizations are due to sepsis. Those with hematological malignancies have a 15-fold increase in severe sepsis compared to the general population. Moreover, the rates of sepsis-related mortality are higher among patients with AML than those with other cancers (30% vs. 21%) [4]. Patients with newly diagnosed or relapsed acute leukemia and lymphomas are also at risk of developing acute kidney injury (AKI). Various factors, such as neutropenia and increased exposure to nephrotoxic antibiotics, may play a role in the pathogenesis of AKI [5]. Additionally, sepsis itself is a common cause of AKI [6]. Furthermore, the presence of AKI may compromise the efficacy of anticancer treatments. The presence of AKI in patients with sepsis also imposes a considerable strain on healthcare resources and leads to extended hospital stays [6,7]. Lahoti et al. reported that 36% of adult AML patients develop AKI during chemotherapy and relatively mild elevations in creatinine are associated with higher mortality [8]. Ballo et al. published that AKI adversely affects the clinical course of AML patients undergoing chemotherapy [9].

Although both sepsis and AKI are associated with increased mortality in AML, there is a lack of data on the outcomes of AML patients with sepsis who develop AKI [7]. Knowledge about the associations related to AKI in leukemias is also limited when compared to malignancies such as multiple myeloma [10]. To address this gap, our study aimed to understand AKI outcomes among hospitalized patients with sepsis and AML. We investigated the effects of AKI on patients with AML and sepsis to assess the incidence of all-cause mortality and other complications among these patients.

## 2. Materials and Methods

### 2.1. Methods

#### 2.1.1. Data Source and Patient Selection

The National Inpatient Sample (NIS) is a comprehensive database of the United States (US) that is a part of the Healthcare Cost and Utilization Project (HCUP). It comprises the discharge data obtained from a 20% stratified sample of the discharges across 48 states and the District of Columbia. The database includes information on hospital stays and was developed through a partnership sponsored by the Agency for Healthcare Research and Quality (AHRQ) [11]. Specifically, data regarding patient demographics, hospital characteristics, insurance, and primary and secondary diagnoses can be obtained.

We utilized the NIS to conduct a retrospective analysis of the 2016–2020 NIS data to identify adult hospitalizations with AKI, sepsis, and AML. The International Classification of Diseases Clinical Modification codes (ICD-10-CM) were used to identify primary and secondary diagnoses [12]. We selected the years 2016–2020 as they represent the most recent available data at the time of analysis while ensuring relevance.

Patients ≥18 years old with diagnoses of sepsis, AML, and AKI were identified and included using ICD-10 codes. End-stage renal disease (ESRD) patients on dialysis were excluded from the analysis. Groups were created based on the presence or absence of AKI. The selection process is shown in Figure 1. Patient demographics, hospital details, insurance, comorbidities, and other characteristics were compared between groups. Additionally, the Elixhauser comorbidity index was used to compare the severity of comorbidities between the study groups [13]. The ICD-10 codes used in the study are shown in Appendix A. Since the NIS contains de-identified information, approval from an Institutional Review Board (IRB) was not required.

#### 2.1.2. Outcome Assessed

The primary outcome was all-cause inpatient mortality. The secondary outcomes were septic shock, fluid and electrolyte vasopressor support, mechanical ventilation, mechanical ventilation > 96 h, and length of stay (LOS) due to AKI in patients with sepsis and AML. We defined fluid and electrolyte disorders as disorders due to hypernatremia, hyponatremia, acidosis, alkalosis, mixed acid–base disorders, hyperkalemia, hypokalemia, and fluid overload.

#### 2.1.3. Statistical Analysis

The continuous variables were presented as the mean and standard deviation (SD), whereas the categorical variables were presented as frequencies and percentages. Student’s *t*-test was used to compare continuous variables between groups, and a chi-square test was used to compare categorical variables. We performed a univariable analysis to calculate the crude odds ratio for factors associated with AKI and the outcomes of patients. Next, a multivariable analysis was conducted to estimate the adjusted odds ratio using the variables that showed an association during univariable analysis and also by incorporating variables that were shown to have an association from prior studies [4,8]. A *p*-value of <0.05 was considered statistically significant. All the statistical analyses were performed using Stata/MP, version 18 (Stata Corp, College Station, TX, USA) [14].

## 3. Results

### 3.1. Patient and Hospital Characteristics

During the years 2016 to 2020, there were a total of 288,435 adult in-patient admissions with sepsis among patients with AML. Of these patients, 61,955 (21.4%) had AKI (shown in Figure 1). Patients with AKI were older, with a mean age of 66.1 years, compared to those without AKI (60.4 years, *p* < 0.001). Patients with AKI between the ages of 60 and 74 years were more likely to be admitted than patients of other ages (44.2%). The rates of other age groups were 18–44 years (18.7%), 45–59 years (17.3%), and 75 years or older (29.8%). There was a higher percentage of males with AKI (63.1%) compared to those without AKI (52.8%, *p* < 0.001). In terms of racial composition, White patients constituted the majority but were similar in distribution among both AKI and non-AKI groups (70.3% with AKI vs. 70.1% without AKI). Black patients were more prevalent in the AKI group than in the non-AKI group (12% vs. 9.2%, *p* < 0.001).

No significant differences were observed among household income quartiles (*p* = 0.632). Geographically, hospitals in the south had a high incidence of sepsis with AKI (36.0%), while the incidence was lowest in the west (19.4%) (*p* = 0.001). Large hospitals had more admissions with sepsis and AML (68.3%) but without significant differences between the AKI and non-AKI groups (*p* = 0.408). Among the types of institutions, urban teaching hospitals had more admissions of patients with AKI when compared to non-AKI patients (87.1% vs. 86.9%, *p* < 0.001). AKI was higher among Medicare beneficiaries (60.0% vs. 47.4% *p* < 0.001). Among microorganisms causing sepsis, Gram-negatives constituted the majority (8.5% vs. 6.2%, *p* < 0.001), but other organisms such as Staphylococcus (Staph) (2.8% vs. 1.7%, *p* < 0.001) and Enterococcus species (2.5% vs. 1.5%, *p* < 0.001) were higher in the AKI group.

Patients with AKI had higher Elixhauser comorbidity index scores ≥ 3, indicating a significant comorbidity burden (*p* < 0.001). Chronic kidney disease (CKD) was notably more prevalent in patients with sepsis and AKI (31.0% vs. 8.7%, *p* < 0.001). Other significant comorbidities in the AKI group included diabetes mellitus (DM) with complications (13.6% vs. 5.3%, *p* < 0.001), congestive heart failure (CHF) (26.6% vs. 12.6%, *p* < 0.001), cardiac arrhythmias (33.4% vs. 20.6%, *p* < 0.001), cerebrovascular disease (6.9% vs. 3.6%, *p* < 0.001), and moderate or severe liver disease (1.8% vs. 0.6%, *p* < 0.001). Tumor lysis syndrome (TLS) occurred more in the AKI group (11.1% vs. 1.4%, *p* < 0.001). The demographics and patient characteristics are shown in Table 1.

### 3.2. Factors Associated with AKI

Among AML patients, several factors were shown to have an association between sepsis and AKI. AKI was less likely in females (aOR: 0.64, 95% CI: 0.61–0.67, *p* < 0.001). Black people had a higher association with AKI (aOR: 1.41, 95% CI: 1.30–1.52, *p* < 0.001). The presence of a higher number of comorbidities showed a statistically significant association with AKI in sepsis in AML patients. On the Elixhauser comorbidity index, when compared to a score of 0, a score of 1 doubled the association of developing AKI (aOR: 2.0, 95% CI: 1.74–2.50, *p* < 0.001), and a ≥5 comorbidity index score increased this by more than 13 times (aOR: 13.7, 95% CI: 11.5–16.2, *p* < 0.001). Among microorganisms, infections due to Staphylococcus (Staph) and Enterococcus species were associated with a higher risk (aOR: 1.52 and 1.55), respectively. Among comorbidities, the presence of chronic kidney disease was associated with a higher risk (aOR: 2.6, 95% CI: 2.45–2.77, *p* < 0.001). An association was also seen with DM with complications (aOR: 1.42, 95% CI: 1.31–1.53, *p* < 0.001), CHF (aOR: 1.26, 95% CI: 1.18–1.33, *p* < 0.001), cerebrovascular disease (aOR: 1.42, 95% CI: 11.29–1.57, *p* < 0.001), and moderate/severe liver disease (aOR: 1.9, 95% CI: 1.6–2.4, *p* < 0.001). A strong correlation was seen between TLS and AKI (aOR: 6.6, 95% CI: 5.9–7.3, *p* < 0.001). Table 2 shows the factors associated with AKI.

Multivariable analysis was performed by adjusting for age, sex, race, median income for zip code, Elixhauser comorbidity index, hospital bed size, hospital location, teaching status, and insurance.

### 3.3. In-Hospital Outcomes

Patients with AKI had a higher risk of fluid and electrolyte disorders (aOR: 2.2, 95% CI: 2.1–2.4, *p* < 0.001), septic shock (aOR: 6.3, 95% CI: 5.7–6.9, *p* < 0.001), vasopressor requirement (aOR: 5, 95% CI: 4.3–5.8, *p* < 0.001), and mechanical ventilation (aOR: 5.2, 95% CI: 4.7–5.7, *p* < 0.001). They also had a higher risk of requiring a longer mechanical ventilation duration of >96 h (aOR: 6.3, 95% CI: 5.4–7.5, *p* < 0.001). The risk of all-cause mortality was high (aOR: 3.8, 95% CI: 3.6–4.1, *p* < 0.001).

The presence of AKI was also associated with a 3.31-day (95% CI: 2.94–3.68, *p* < 0.001) increase in LOS, which was statistically significant. The mean length of stay (LOS) for patients with sepsis and AKI was 15.4 days compared to 10.8 days for those without AKI (*p* < 0.001). The in-hospital outcomes are detailed in Table 3.

Multivariable analysis was performed by adjusting for age, sex, race, hospital bed size, hospital location and teaching status, insurance, staphylococcus, streptococcal, enterococcus, Gram-negative infections, hypertension, diabetes mellitus, diabetes mellitus with complications, hyperlipidemia, coronary artery disease, cardiac arrhythmias, peripheral vascular disease, cerebrovascular disease, chronic kidney disease, fluid and electrolyte disorders, chronic obstructive pulmonary disease, moderate–severe liver disease, rheumatological diseases, anemia, obesity, and smoking.

### 3.4. Discussion

Our analysis of patients with sepsis and AML revealed several key findings. The incidence of AML-related AKI increased significantly among males and Black individuals. A higher comorbidity burden and the presence of CKD was associated with AKI. There were trends of increased odds of inpatient mortality and complications such as fluid and electrolyte disorders, septic shock, increased vasopressor support, need for mechanical ventilation, and prolonged ventilator requirements in the AKI group. These complications also contributed to more extended hospital LOS.

The pathogenesis of AKI in cancer patients with sepsis involves hemodynamic changes, inflammatory cascades, and microcirculatory dysfunction. Hemodynamic changes lead to reduced renal perfusion, while the inflammatory responses cause direct tubular injury [7,15]. The development of AKI in septic patients with hematological malignancies also shares a similar interplay of hemodynamic and molecular events [16].

Multiple AML-specific risk factors increase the risk of developing AKI, in addition to sepsis. Patients with AML are at risk for developing tumor lysis syndrome, especially when their disease is not in remission. Tumor lysis syndrome can cause electrolyte abnormalities, metabolic acidosis, and acute tubular necrosis (ATN). AML patients can also develop lysozyme nephropathy, a proteinuric condition that can cause AKI and ATN. In addition to these, AML can cause AKI due to direct kidney infiltration and leukostasis. Glomerular lesions have also been implicated in renal impairment in AML patients [17,18].

Several associations were noted in our study. Among hospitalized patients with sepsis and AML, the rates of AKI were high among males aged 60–74 years. These findings are akin to those of Lahoti et al., who noted that increasing age, >55 years, was a risk factor in their cohort of AML patients hospitalized for various reasons. In addition, they identified several predictors of AKI among patients with AML and high-risk myelodysplasias, such as leukopenia, hypoalbuminemia, mechanical ventilation, vasopressors, nephrotoxic antibiotics, and chemotherapy. It was also shown that patients who achieved complete cancer remission had a favorable outcome regardless of whether renal failure was present [8].

A retrospective study conducted by Ballo et al. showed that AML patients with AKI had increased mortality, and the risk for mortality increased proportionately to the severity of AKI. AML patients with AKI had more days with fever, prolonged ICU stay, lower albumin levels, and higher procalcitonin levels. Male patients, age > 60, adverse risk AML, hypoalbuminemia, and patients on ACE inhibitors and liposomal amphotericin B treatment were at an increased of developing AKI [9]

Our present study found that comorbidities such as DM with complications, CHF, cerebrovascular disease, CKD, and moderate or severe liver disease were associated with AKI among patients with sepsis and AML. The Elixhauser comorbidity index assigns weights to comorbidities, creating a score that could be used clinically to identify high-risk populations. High Elixhauser comorbidity index scores correlate with increased mortality risk and rehospitalization risk [13]. We found that high Elixhauser comorbidity scores were associated with the occurrence of AKI. Tumor lysis syndrome (TLS) is a common cause of AKI in AML patients, which is usually not present in other malignancies. The release of intracellular contents, such as potassium, phosphate, and uric acid, into the bloodstream leads to tubular dysfunction [16]. TLS was present in about 11% of our AKI cohort and was also associated with a six-fold increase in AKI. Staph, Enterococcus, and Gram-negative infections were higher in the AKI group. Gram-negative infections occurred more than infections due to other microorganisms.

However, despite the higher percentage of patients aged 60–74 years with AKI, the multivariable analysis showed only a small effect size, suggesting the possibility that other factors may play a role in causing this age group to develop more AKI. In our study, Black people had an increased risk for AKI. The heightened prevalence of arterial hypertension, DM, chronic kidney disease (CKD), and limited healthcare accessibility are the likely causes for the elevated susceptibility of Black individuals to develop AKI [19].

Shum et al. reported that patients with sepsis and AKI had higher APACHE IV (Acute Physiology and Chronic Health Evaluation IV) scores when compared to patients with AKI without sepsis, indicating a higher disease severity. In their cohort of patients with sepsis, vasopressor and renal replacement therapy (RRT) requirements were more commonly seen in the AKI group. The mortality rates at 3 months were similar between septic and non-septic AKI in their study [6]. However, this is not the case in patients with cancers. The mortality rates are higher in sepsis in the presence of a malignancy. Yang et al. assessed 356 patients with sepsis and various malignancies who were admitted to the ICU. In the AKI group, ventilator and RRT requirements were significantly higher than those in the non-septic AKI group, and there was a notable increase in the 28-day mortality [15]. High mortality was also shown in another study by Malik et al. Patients with sepsis and AML had higher mortality rates than other cancers, with rates close to 30%. Among these patients, those with AML relapse had a higher risk of death during hospitalization than those in remission [4]. In our analysis, the incidence of AKI was also high among patients who were not in remission and was less among patients in remission. The mortality rate in the AKI group was 22.7% in our study, which was less than the study by Malik et al.

A strong correlation was also noted between mechanical ventilation and diuretics with AKI in the study by Lahoti et al. The utilization of mechanical ventilation and diuretic medication could arise due to the excessive accumulation of fluids during the initial phases of chemotherapy. A positive fluid balance has been linked to higher mortality rates in sepsis [8]. Our retrospective data analyses also demonstrated that the presence of AKI was associated with a higher risk of vasopressor and ventilator requirements. The risk of more extended mechanical ventilation requirement was about six-fold higher in the AKI group in our dataset, which was not shown in the other studies.

AKI due to sepsis has significant long-term consequences. Patients who developed an episode of AKI alongside newly diagnosed acute leukemia or lymphoma had a 1.9-fold risk of experiencing a substantial loss in estimated glomerular filtration rate (eGFR). These patients were also more likely to develop chronic kidney disease (CKD) even if they had normal kidney function at the beginning [5]. The majority of chemotherapeutic drugs utilized undergo renal clearance; thus, AKI and CKD preclude them from treatment options. Furthermore, there is also an elevated risk of cardiac complications among hospitalized AML patients with CKD [20,21]. It is also known from prior studies that AKI in patients with pre-existing CKD causes a higher risk for the requirement of renal replacement therapy (RRT), which may be long-term [22]. While the risk of mortality due to AKI requiring RRT has decreased in recent years, there is an increasing trend in the incidence of AKI requiring RRT due to sepsis [23].

We also demonstrate that the presence of AKI prolongs hospitalization, resulting in increased LOS. The increase in LOS was around 42.5% in the presence of AKI. A previous study showed that RRT requirements increased hospital costs by 21% in all malignancies [24]. Moreover, in our analysis, when compared to non-AKI dispositions, AKI patients had fewer discharges to home (34.6 vs. 61.6%) and more discharges with home health care (22.7 vs. 7.2%), thus showing increased resource utilization. However, for unclear reasons, discharges to skilled nursing facilities or long-term acute care hospitals were less (14.6 vs. 21.3%).

### 3.5. Limitations

Our study has certain limitations. The NIS database analysis is retrospective, albeit with a large dataset allowing a population-level analysis. The accuracy of the data relies on the accuracy of the ICD-10 codes entered by providers. There is also the possibility of the presence of misclassification bias. Many definitions exist for AKI, but the most widely used ones are Kidney Disease Improving Global Outcomes (KDIGO); Acute Kidney Injury Network (AKIN); and the Risk, Injury, Failure, Loss of kidney function, and End stage kidney disease (RIFLE) [25,26]. There may be non-uniformity in the definitions used to diagnose AKI, and subclinical AKI could have been missed. Microbiology data are insufficient in retrospective investigations based on NIS and have not been validated [27]. Patients with electrolyte disorders were isolated based on ICD codes, but this may include patients who had electrolyte disorders due to mechanical ventilation. Similarly, laboratory data and medication details are not available in the NIS. Due to the retrospective nature of the database analysis and the limitations of the ICD codes, we were unable to find the proportion of patients who developed AKI due to nephrotoxic medications. Thus, there may be some overlap on the causal effects while using the NIS database. Another drawback of the database is the lack of longitudinal follow up. Nonetheless, the NIS dataset allows for large analyses without possible selection bias. In addition, we conducted the study by incorporating many confounders that could potentially influence the multivariable analysis.

## 4. Conclusions

To our knowledge, this is the first nationwide, population-based study to describe the associations and outcomes of AKI in patients with sepsis and acute myeloid leukemia (AML). AKI was associated with increased mortality and complications. Resource utilization, such as LOS and home health care requirements, were significant for these patients. Randomized control trials are required to examine potential risk factors for AKI and factors involved in RRT requirements. Timely identification and risk stratification can potentially reduce the number of deaths and disease-related complications.

## Figures and Tables

**Figure 1 jcm-14-02243-f001:**
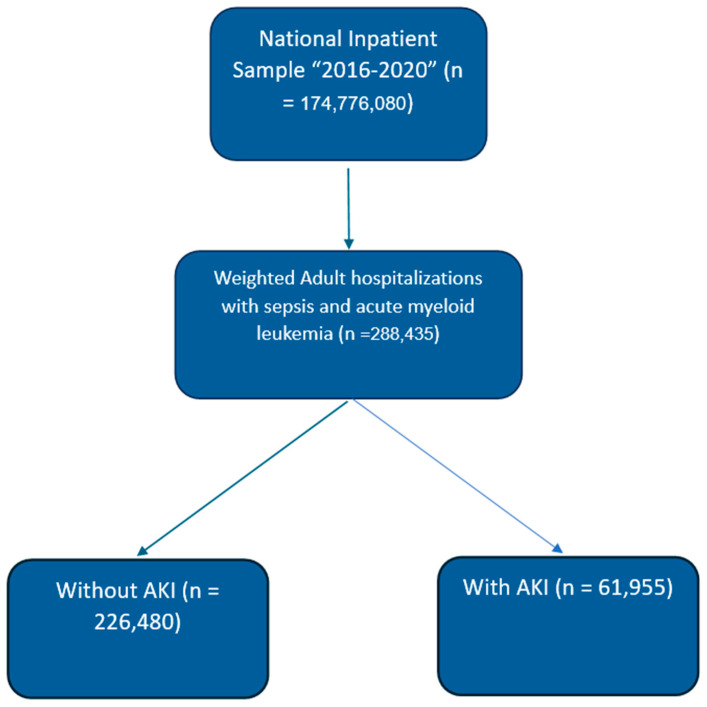
Patient selection (weighted flow chart). Abbreviations: AKI—acute kidney injury.

**Table 1 jcm-14-02243-t001:** Weighted patient characteristics and comorbidities of adult hospitalizations with sepsis, acute myeloid leukemia (AML), and acute kidney injury (AKI) from the NIS between 2016 and 2020.

	Characteristics	Sepsis in AML Without AKI, n = 226,480	Sepsis and AML with AKI, n = 61,955 (21.4%)	*p*-Value
	Age in years, Mean ± SD	60.4 ± 16.8	66.15 ± 14.47	<0.001
Age groups				<0.001
18–44 yrs		39,535 (17.6)	5415 (8.7)	
45–59 yrs		50,925 (22.6)	10,715 (17.3)	
60–74 yrs		88,420 (39.1)	27,365 (44.2)	
75 yrs or older		46,580 (20.7)	18,460 (29.8)	
Gender				<0.001
Female		106,915 (47.2)	22,910 (36.9)	
Male		119,565 (52.8)	39,045 (63.1)	
Race				<0.001
White		158,825 (70.1)	43,605 (70.3)	
African American		20,965 (9.2)	7455 (12)	
Hispanic		212,75 (9.4)	4270 (6.8)	
Median household income for patient’s zip code				0.632
Quartile 1		51,370 (22.6)	14,000 (22.5)	
Quartile 2		55,715 (24.7)	15,100 (24.4)	
Quartile 3		57,715 (25.4)	15,630 (25.3)	
Quartile 4		57,840 (25.6)	16,225 (26.2)	
Hospital region				0.001
Northeast		44,660 (19.8)	13,050 (21)	
Midwest		51,095 (22.5)	14,570 (23.6)	
South		83,830 (37)	22,310 (36)	
West		46,895 (20.7)	12,025 (19.4)	
Hospital size				0.408
Small		26,895 (11.9)	7605 (12.3)	
Medium		45,030 (19.9)	12,055 (19.4)	
Large		154,555 (68.2)	42,295 (68.3)	
Hospital location and teaching status				<0.001
Rural		7075 (3.2)	1475 (2.3)	
Urban non-teaching		22,505 (9.9)	6540 (10.6)	
Urban teaching		196,900 (86.9)	53,940 (87.1)	
Medical Insurance				<0.001
Medicare		107,380 (47.4)	37,215 (60)	
Medicaid		28,700 (12.7)	5415 (8.7)	
Private		78,385 (34.6)	16,580 (26.8)	
No insurance		4495 (1.9)	935 (1.5)	
Elixhauser comorbidity Index				<0.001
0		22,320 (9.8)	905 (1.4)	
1		39,205 (17.3)	3480 (5.6)	
2		47,160 (20.8)	7345 (11.8)	
3–4		43,195 (19.1)	22,435 (36.2)	
≥5		74,600 (33)	27,790 (36.7)	
Microorganism				
Staphylococcus		3975 (1.7)	1740 (2.8)	<0.001
Streptococcus		3845 (1.6)	1045 (1.6)	0.629
Enterococcus		3455 (1.5)	1605 (2.5)	0.001
Gram-negative		14,215 (6.2)	5280 (8.5)	0.002
AML status				
AML not in remission		153,500 (67.6)	44,920 (72.4)	<0.001
AML in relapse		32,465 (14.3)	9140 (14.5)	0.241
AML in remission		41,405 (18.1)	8175 (13.1)	<0.001
Comorbidities				
Arterial hypertension		95,585 (42.2)	23,190 (37.4)	<0.001
Diabetes mellitus		32,445 (14.3)	8480 (13.8)	0.056
Diabetes mellitus with complications		12,035 (5.3)	8435 (13.6)	<0.001
Hyperlipidemia		63,045 (27.8)	20,280 (32.7)	<0.001
Congestive Heart Failure		28,545 (12.6)	16,525 (26.6)	<0.001
Peripheral vascular disease		10,330 (4.6)	3915 (6.3)	<0.001
Coronary artery disease		31,365 (13.8)	12,115 (19.5)	<0.001
Cardiac arrhythmias		46,645 (20.6)	20,710 (33.4)	<0.001
Cerebrovascular disease		8235 (3.6)	4310 (6.9)	<0.001
Chronic kidney disease		19,824 (8.7)	19,230 (31)	<0.001
Chronic obstructive pulmonary disease		35,015 (15.4)	10,750 (17.3)	<0.001
Moderate/severe liver disease		1420 (0.6)	1145 (1.8)	<0.001
Rheumatological disorders		5930 (2.6)	1715 (2.7)	0.401
Anemia		7525 (3.3)	2725 (4.4)	<0.001
Dementia		3765 (1.6)	1655 (2.7)	<0.001
Obesity		21,990 (9.7)	6910 (11.1)	<0.001
Smoking		19,060 (8.4)	4020 (6.4)	<0.001
Other factors				
Tumor lysis		3220 (1.4)	6915 (11.1)	<0.001
Disposition				<0.001
Discharge home		139,640 (61.6)	21,450 (34.6)	
Discharge to HHC		16,290 (7.2)	14,075 (22.7)	
Discharge to SNF, ICF, LTACH		48,405 (21.3)	9075 (14.6)	
LOS, mean ± SD		10.8 ± 13.1	15.41 ± 18	<0.001

Abbreviations: HHC, home with health care; ICF, intermediate care facility; LTACH, long-term acute care hospital; NIS, national inpatient sample; SD, standard deviation; SNF, skilled nursing facility.

**Table 2 jcm-14-02243-t002:** Factors associated with acute kidney injury (AKI) in sepsis and acute myeloid leukemia (AML).

Variable		Multivariable Adjusted OR (95% CI)	*p*-Value
Age		1.01 (1.0–1.01)	<0.001
Sex	Female	0.64 (0.61–0.67)	<0.001
Race (reference, White)			
	Black	1.41 (1.30–1.52)	<0.001
	Hispanic	0.98 (0.89–1.07)	<0.001
Median household income for patient’s zip code % (reference, quartile 1)			
	Quartile 2	1.038 (0.97–1.10)	0.262
	Quartile 3	1.039 (0.970–1.112)	0.270
	Quartile 4	1.04 (0.973–1.118)	0.232
Elixhauser comorbidity index (reference, 0)			
	1	2.0 (1.74–2.50)	<0.001
	2	3.46 (2.91–4.12)	<0.001
	3–4	6.56 (5.54–7.78)	<0.001
	≥5	13.72 (11.55–16.29)	<0.001
Region (reference, Northeast)			
	Midwest	0.90 (0.83–0.98)	0.014
	South	0.89 (0.83–0.96)	0.004
	West	0.90 (0.83–0.97)	0.014
Hospital bed size (reference, small)			
	Medium hospital	0.96 (0.88–1.05)	0.402
	Large hospital	1.10 (1.02–1.19)	0.009
Hospital location and teaching status (reference, rural hospital non-academic)			
	Urban non-teaching	1.45 (1.23–1.69)	<0.001
	Urban teaching	1.58 (1.37–1.89)	<0.001
Insurance (reference, Medicare)			
	Medicaid	0.97 (0.88–1.07)	0.628
	Private	1.03 (0.97–1.09)	0.322
	No insurance	1.07 (0.90–1.29)	0.408
Organism			
	Staphylococcus	1.52 (1.31–1.75)	<0.001
	Streptococcus	1.22 (1.02–1.46)	0.027
	Enterococcus	1.55 (1.34–1.81)	<0.001
	Gram-negative	1.32 (1.21–1.44)	<0.001
Comorbidities			
	Arterial hypertension	0.52 (0.50–0.55)	<0.001
	Diabetes mellitus	0.60 (0.56–0.64)	<0.001
	Diabetes mellitus with complications	1.42 (1.31–1.53)	<0.001
	Hyperlipidemia	0.78 (0.74–0.82)	<0.001
	Congestive heart failure	1.26 (1.18–1.33)	<0.001
	Peripheral vascular disease	0.73 (0.66–0.80)	<0.001
	Coronary artery disease	0.82 (0.77–0.87)	<0.001
	Cardiac arrhythmias	1.04 (0.99–1.10)	0.108
	Cerebrovascular disease	1.42 (1.29–1.57)	<0.001
	Chronic kidney disease	2.60 (2.45–2.77)	<0.001
	Chronic obstructive pulmonary disease	0.66 (0.62–0.70)	<0.001
	Moderate/severe liver disease	1.99 (1.64–2.40)	<0.001
	Rheumatological disorders	0.71 (0.62–0.81)	<0.001
	Anemia	0.82 (0.73–0.91)	<0.001
	Dementia	1.02 (0.88–1.17)	0.785
	Obesity	0.79 (0.73–0.85)	<0.001
	Smoker	0.68 (0.62–0.75)	<0.001
Other factors			
	Tumor lysis	6.61 (5.95–7.35)	<0.001

**Table 3 jcm-14-02243-t003:** Outcomes of patients with acute kidney injury (AKI), sepsis, and acute myeloid leukemia (AML).

Outcomes	AKI in Sepsis and AML, N	Multivariable Adjusted OR (95% CI)	*p*-Value
Length of stay (days)	15.41 ± 18	3.31 (2.94–3.68)	<0.001
Fluid and electrolyte disorders	36,335 (58.6%)	2.29 (2.18–2.41)	<0.001
Septic shock	11,575 (18.6%)	6.31 (5.77–6.90)	<0.001
Vasopressor requirement	34,995 (56.4%)	5.0 (4.32–5.85)	<0.001
Mechanical ventilation	9485 (15.3%)	5.29 (4.79–5.79)	<0.001
Mechanical ventilation >96 h	3800 (6.1%)	6.39 (5.44–7.51)	<0.001
All-cause mortality	14,100 (22.7%)	3.87 (3.60–4.16)	<0.001

## Data Availability

The National Inpatient Sample data are publicly available at the HCUP-US Home Page (https://hcup-us.ahrq.gov/databases.jsp (accessed on 12 April 2024)).

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
