# Peer review of "Outcomes of Acute Kidney Injury Among Hospitalized Patients with Sepsis and Acute Myeloid Leukemia: A National Inpatient Sample Analysis"

_jcm, 2025, doi:10.3390/jcm14072243_

Round 1
Reviewer 1 Report
Comments and Suggestions for Authors
Materials and Methods: What does 20% stratified mean? Maybe it is a language issue, but I do not understand that part of the database. Please rephrase.
Why were the years 2016 to 2020 chosen?
Figure 1= There Is no number presented of included patients.
How was made sure that the “electrolyte disorders” did not just result from mechanical ventilation?
You write:” Among age groups, patients with AKI between 60-74 years had higher admission rates than others (44.2%)” Does this account for the demographics in the general population/AML patients? If not, why?
What percentage of patients was treated in ICU?
In your opinion, what conclusions can be drawn from the data on microorganisms you presented?
How do you explain the AKI protective effect of these vascular risk factors, e.g. hypertension? (table 2). Adding to this, later on you write “the heightened prevalence of arterial hypertension, DM,
chronic kidney disease (CKD), and limited healthcare accessibility are the likely causes for
the elevated susceptibility of Black individuals to develop AKI”. However hypertension seems to be protective in your data.
You present a multivariable analysis in table 2. Why did you not further include further variables presented in table 2 for a further analysis? I believe there is a great deal of overlap between the factors and I would be interested in adjusted risk factors.
You mention in the introduction that data for this specific question is scarce. As clearly there has been some previous risk and outcome data on AKI and AKI in sepsis, please explain how AML patient differ to “normal” ones. Are there AML specific risks? If so, why were they not included in the analysis, except for TLS? (thinking about AML medication for example)
To me it reads like the AKI leads to worse outcomes, which might be true, but cannot be inferred from your retrospective data. This should be mentioned.
In the discussion section, to be it is not always clear which statements are inferred from the data presented here and which are based on other studies. Please improve on the phrasing and/or restructure the discussion.
“Many definitions exist for AKI, but the most widely used ones are”. The sentence is not finished.
Author Response
We thank the reviewers for their time and valuable comments and suggestions. We have revised the manuscript as per the suggestions.
Reviewer: 1
Comments to the Author
Materials and Methods: What does 20% stratified mean? Maybe it is a language issue, but I do not understand that part of the database. Please rephrase.
Reply: National Inpatient Sample (NIS) is a sample of approximately 20% of all U.S. hospital discharges and this is used as a true representation of the actual inpatient hospital discharges. The sentence has been modified in the manuscript.
Why were the years 2016 to 2020 chosen?
Reply: The 2016-2020 period was chosen as it was the most recent available data at the time of analysis, we have added this to the manuscript.
Figure 1= There Is no number presented of included patients.
Reply: It is 288,435.
How was made sure that the “electrolyte disorders” did not just result from mechanical ventilation?
Reply: We acknowledge that mechanical ventilation can contribute to electrolyte disturbances. We have added this as a limitation of our study.
You write:” Among age groups, patients with AKI between 60-74 years had higher admission rates than others (44.2%).” Does this account for the demographics in the general population/AML patients? If not, why?
Reply: Thank you for the comment. We were referring to the 60-74-year-old AML patients with sepsis group. Multiple retrospective studies done on inpatient sepsis data samples showed increased hospitalization rates among patients >65 years of age compared to 18-64 years of age. We could not find a data set that stratified the sample into 60-74 and > 75 age categories among the sepsis hospitalization rates in the general population.
Overview of Outcomes for Inpatient Stays Involving Sepsis, 2016–2021
What percentage of patients was treated in ICU?
Reply: The National Inpatient Sample database does not provide this information, so we were unable to determine this and do further analysis. We have added this as a limitation.
In your opinion, what conclusions can be drawn from the data on microorganisms you presented?
Reply: Thank you for the comment. We already mentioned that gram-negative organisms accounted for the majority of infections in these patients. We also concluded that a higher trend was noted with staph aureus and enterococcal infections.
How do you explain the AKI protective effect of these vascular risk factors, e.g. hypertension? (table 2). Adding to this, later on you write “the heightened prevalence of arterial hypertension, DM, chronic kidney disease (CKD), and limited healthcare accessibility are the likely causes for the elevated susceptibility of Black individuals to develop AKI”. However hypertension seems to be protective in your data.
Reply:
Thank you for the comment. In our retrospective sample population, we saw a trend of decreased AKI risk concerning certain comorbidities. Hypertension was one of the comorbidities that showed a trend of decreased risk, but we are not sure about the exact reason for those results. We do not think hypertension would confer any protection against the development of AKI in AML patients with sepsis. We updated our manuscript regarding the potential causes of increased AKI risk among African American patients.
You present a multivariable analysis in table 2. Why did you not further include further variables presented in table 2 for a further analysis? I believe there is a great deal of overlap between the factors and I would be interested in adjusted risk factors.
Reply: We believe that we have taken into account all the factors that showed statistical significance to see if there was any association between various comorbidities and AKI in Table 2. We have added the factors that showed statistical significance to find out if there is an association with the outcomes and this was presented in Table 3
You mention in the introduction that data for this specific question is scarce. As clearly there has been some previous risk and outcome data on AKI and AKI in sepsis, please explain how AML patient differ to “normal” ones. Are there AML specific risks? If so, why were they not included in the analysis, except for TLS? (thinking about AML medication for example)
Reply: The pathogenesis of AKI in cancer patients with sepsis involves hemodynamic changes, inflammatory cascades, and microcirculatory dysfunction. The development of AKI in septic patients with hematological malignancies also shares a similar interplay of hemodynamic and molecular events. In addition to these, AML can cause AKI due to direct kidney infiltration and leukostasis. Glomerular lesions have also been implicated in renal impairment in AML patients. This has been explained in the discussion section.
We recognize the possible impact of nephrotoxic chemotherapy on AKI but NIS database does not provide medication details. We have added this as a limitation.
To me it reads like the AKI leads to worse outcomes, which might be true, but cannot be inferred from your retrospective data. This should be mentioned.
Reply: Thank you for your comment. We agree that definitive conclusions should not be drawn based on our retrospective data. We updated our manuscript to suggest trends rather than draw definitive conclusions.
In the discussion section, to be it is not always clear which statements are inferred from the data presented here and which are based on other studies. Please improve on the phrasing and/or restructure the discussion.
Reply: Thank you for the comment. We updated our discussion and tried to improve the clarity.
“Many definitions exist for AKI, but the most widely used ones are”. The sentence is not finished.
Reply: Thank you for pointing this out. We have added the remaining.
Reviewer 2 Report
Comments and Suggestions for Authors
This is a high-quality analysis of the outcomes of acute kidney injury (AKI) in patients with acute myeloid leukemia (AML) and sepsis using a national inpatient sample. The study addresses an important clinical issue with a robust methodology and clear results. Here are a few comments on areas needing further clarification and discussion.
Main concerns:
- Although this is the first study on AML patients with sepsis, the finding that AKI increases mortality is well-known in sepsis cases. The authors should acknowledge this early and then explore how AML-specific factors complicate AKI in sepsis in the Discussion section.
- The analysis seems to overlook the potential impact of nephrotoxic chemotherapy, which is a significant factor in AKI among AML patients. Including this variable could provide a more comprehensive understanding of AKI risk in this population.
Minor points:
- Please add the AKI definition used.
- How were confounding factors handled in the multivariable analysis? Were interaction terms considered?
- Consider adding a figure illustrating key findings such as Kaplan-Meier survival curves.
Author Response
Reviewer: 2
Comments to the Author
Although this is the first study on AML patients with sepsis, the finding that AKI increases mortality is well-known in sepsis cases. The authors should acknowledge this early and then explore how AML-specific factors complicate AKI in sepsis in the Discussion section.
Reply: Thank you for the comment. We acknowledged that the AKI increases mortality in sepsis patients. We added a few sentences in the discussion about how certain AML-specific risk factors increase the risk for AKI and sepsis.
The analysis seems to overlook the potential impact of nephrotoxic chemotherapy, which is a significant factor in AKI among AML patients. Including this variable could provide a more comprehensive understanding of AKI risk in this population.
Reply: We recognize the possible impact of nephrotoxic chemotherapy on AKI but NIS database does not provide medication details. We have added this as a limitation.
Please add the AKI definition used.
Reply: Thank you for your comment. The NIS database lacks the variables to set a pre-specified definition. We relied on the data provided by the providers and the definitions they used to diagnose AKI. We have added this as a limitation of or study and also added that the accuracy of the data relies on the accuracy of the ICD-10 codes entered by providers.
How were confounding factors handled in the multivariable analysis? Were interaction terms considered?
Reply: Thank you for your question. We initially performed a univariable analysis to calculate the crude odds ratio for factors associated with AKI and the outcomes of patients. After this, a multivariable analysis was conducted to estimate the adjusted odds ratio using the variables that showed a statistically significant association during univariable analysis and also by incorporating variables that were shown to have an association from prior studies. We have mentioned this in the manuscript.
Consider adding a figure illustrating key findings such as Kaplan-Meier survival curves.
Reply: Since NIS lacks longitudinal follow-up, Kaplan-Meier analysis was not feasible. We have added this as a limitation.
Round 2
Reviewer 1 Report
Comments and Suggestions for Authors
I believe you have adressed my comments adequately in the revised verison of the manuscript. Some improvements have not been made due to the nature of data provided in the database used. This has been adressed in the limitations, however it still lowers the quality of the research. I believe this is scientifically valid.
Author Response
- Extensive english editing - has been done
2. Modifying the discussion. - has been modified as requested.
all the changes have been highlighted in the document